# Antimicrobial Properties of Chitosan and Chitosan Derivatives in the Treatment of Enteric Infections

**DOI:** 10.3390/molecules26237136

**Published:** 2021-11-25

**Authors:** Dazhong Yan, Yanzhen Li, Yinling Liu, Na Li, Xue Zhang, Chen Yan

**Affiliations:** The Affiliated Hospital of North Sichuan Medical College, Nanchong 637000, China; docyandazhong@163.com (D.Y.); yanzhen_li91@sina.com (Y.L.); yinling_liu@hotmail.com (Y.L.); na_li_huludao@163.com (N.L.); xue-zhang@outlook.com (X.Z.)

**Keywords:** chitosan, antimicrobial, antibacterial, antifungal, enteric infection, antimicrobial mechanisms of chitosan, gut microbiota, colonization resistance, drug delivery

## Abstract

Antibiotics played an important role in controlling the development of enteric infection. However, the emergence of antibiotic resistance and gut dysbiosis led to a growing interest in the use of natural antimicrobial agents as alternatives for therapy and disinfection. Chitosan is a nontoxic natural antimicrobial polymer and is approved by GRAS (Generally Recognized as Safe by the United States Food and Drug Administration). Chitosan and chitosan derivatives can kill microbes by neutralizing negative charges on the microbial surface. Besides, chemical modifications give chitosan derivatives better water solubility and antimicrobial property. This review gives an overview of the preparation of chitosan, its derivatives, and the conjugates with other polymers and nanoparticles with better antimicrobial properties, explains the direct and indirect mechanisms of action of chitosan, and summarizes current treatment for enteric infections as well as the role of chitosan and chitosan derivatives in the antimicrobial agents in enteric infections. Finally, we suggested future directions for further research to improve the treatment of enteric infections and to develop more useful chitosan derivatives and conjugates.

## 1. Introduction

Ingestion of pathogens and disruption of normal microbiota cause enteric infections [1,2]. Enteric infections mainly manifest as fairly distinct clinical syndromes, including acute vomiting, acute watery diarrhea, profuse watery diarrhea, invasive or bloody diarrhea (dysentery), persistent diarrhea, and enteric fever [3]. Enteric infections are induced by viruses, bacteria, protozoa, or parasites, e.g., norovirus, *Shigella* spp., *Vibrio cholerae*, *Listeria*, *Shiga* toxin-producing *Escherichia coli* (STEC), *Clostridium difficile*, *Salmonella typhimurium*, and *Giardia lamblia* [4,5]. Pathogens contain virulence factors such as enterotoxins and flagella, which increase intestinal intracellular cyclic nucleotides and activate Cl^−^ channels in the apical membrane of enterocytes, resulting in increased fluid secretion and decreased fluid absorption [6]. Such a mechanism explains the cause of microbial diarrhea.

Rehydration is the backbone of the treatment of enteric infections. Numerous cases show that oral rehydration salts (ORS) can effectively rehydrate patients [7]. Severe sufferers require intravenous fluids and antimicrobial therapy [7]. Antibiotics demonstrated benefits in randomized controlled trials (RCT) in the treatment of infection with *Shigella* [8], *Vibrio cholerae* [9], and Enterotoxigenic *E. coli* (ETEC) [10], especially in moderate or severe cases. Recently, there is a consensus that antibiotics overuse contributed to increased drug resistance and the resulting dysbiosis [2]. Novel antibiotics are massively produced, but the continuous fear of the resulting antibiotic resistance and dysbiosis elicited researchers to concentrate on the application of nonantibiotic compounds as antimicrobial agents [1].

Chitin is a biocompatible and biodegradable polysaccharide extracted from crustaceans, fungi, and insects [11]. Chitin can be converted into its deacetylated derivative, chitosan [1]. Conversion process includes enzymatic and chemical conversions; however, lower cost of chemical conversion contributes to its dominance in mass production in chitosan extraction [12]. In the chemical deacetylation, high NaOH concentrations (50–60%) are used at above 80 °C in the treatment of chitin. Under the most drastic conditions, high NaOH concentrations (50–60%) and high temperatures (130–150 °C) shorten the deacetylation time to less than 2 h [13]. Chitosan, a promising natural polymer possessing antimicrobial properties, shows bactericidal activity. Furthermore, antimicrobial properties, biocompatibility, and nontoxicity made chitosan an ideal compound in medical science [1,14]. The United States Food and Drug Administration (FDA) approved that chitosan is GRAS (Generally Recognized as Safe by FDA) [15,16]. Also, a variety of antimicrobial dressings and drug vehicles using chitosan were approved by the FDA [17].

Positive charges from amino groups of chitosan interact electrostatically with negatively charged components on the microbial membrane, creating antimicrobial properties. The antimicrobial properties are primarily confined to a pH below 6 [18]. This behavior can limit applications and investigations under neutral and alkaline conditions; chemical modification of functional groups can give chitosan derivatives enhanced bioactivity and improved aqueous solubility, which maintains the inherent properties of chitosan as well as magnifies the scope of application [18]. Furthermore, chitosan’s mucoadhesive properties and a capacity to promptly open epithelial tight junctions were exploited in drug delivery across various epithelia [19,20,21,22,23,24]. Apart from the above features, biodegradability and biocompatibility add to chitosan usage as an antimicrobial agent and drug delivery vehicle. Recently, chitosan derivatives were prepared as novel materials to expand the scope of application and drug absorption rates, e.g., nanoparticles, microspheres, and polyelectrolyte complexes [25]. Those materials can be utilized in intestine-targeted drug delivery by protecting drugs from being absorbed and degraded in the upper gastrointestinal tract [26]. As chitosan derivatives can transiently open epithelial tight junctions to facilitate the transmembrane transport of drugs, chitosan derivatives can serve as adjuvants or delivery vehicles for mucosal vaccines [27].

Chitosan can serve as prebiotics to increase the colonic mucosal populations of beneficial bacteria and decrease proinflammatory bacteria, improving imbalanced gut microbiota and mucosal inflammation [28]. Also, it gained widespread popularity as a dietary supplement due to the lipid-lowering effects and anti-diabetic properties in the absence of adverse reactions [11]. Besides, healthy gut microbiota can occupy binding sites in the mucus layer covering enterocytes and prevent pathogens from colonizing the human intestine [29]. Infection with conditional pathogens can occur in dysregulated gut microbiota [2]. Therefore, chitosan can be considered a potential and promising prebiotic to improve gut microbiota against colonization of pathogens.

Although there is an increasing interest in the antimicrobial properties of chitosan and chitosan derivatives were reported [30,31,32], there was no agreement on the role of chitosan in antimicrobial agents or therapy in the treatment of enteric infections. There are limited reviews to summarize and analyze mechanisms directly or indirectly related to antimicrobial properties against enteric pathogens.

This review focuses on the preparation of chitosan derivatives with better antimicrobial properties; additionally, this review illustrates the common mechanism of chitosan with antimicrobial properties in four aspects: positively charged chitosan can disrupt the cell membrane or cell wall; chitosan can form a film on the porins of the cell surface to block the exchange of nutrients, leading to microbial cell death; chitosan can penetrate the cell wall to affect DNA/RNA and protein synthesis; unprotonated amino groups of chitosan can chelate metal ions on the cell surface to disrupt cell walls or membranes. However, some mechanisms are contrasted by some investigations, which indicates that more investigations are required to find out the action mode of antimicrobial activity of chitosan and its derivatives. Furthermore, this review elucidates indirect mechanisms of chitosan, namely, affecting biofilm and improving gut microbiota to enhance colonization resistance against pathogens, and it summarizes the current treatment for enteric infections and the role of chitosan derivatives in the antimicrobial agents in enteric infections. We hope that this review can provide helpful suggestions for the clinical treatment of enteric infections and expand the potential application of chitosan and chitosan derivatives.

## 2. General Properties of Chitosan

Chitosan is a random copolymer (Figure 1), composed of d-glucosamine (the deacetylated ones) and *N*-acetyl-d-glucosamine units linked through β-(1−4) glycosidic linkages [18]. Chitosan can be obtained through partial deacetylation of chitin, which is the primary commercial source of chitosan [33]. Chitin is the second most abundant natural polysaccharide following cellulose [18]. The proportion of the deacetylated glucosamine units defines the deacetylation degree (DD). A DD value from 50 to 100% illustrates that the polymer consists of more d-glucosamine units, implying the generation of chitosan. A DD value from 0 to 50% indicates that the polymer comprises more *N*-acetyl-d-glucosamine units, which indicates the copolymer is still chitin [33]. Functional groups on the chitosan molecules consist of three nucleophilic functional groups (C3–OH, C6–OH, and C2–NH_2_), acetyl amino, and glycoside bonds (Figure 1). Among them, the stable acetyl amino bond and the glycosidic bond are not easy to modify; the enlarged steric hindrance of C3–OH decreases chemical reactivity significantly [18]. Thus, C6–OH and C2–NH_2_ groups are more likely to introduce other groups to chitosan derivatives to improve the physical and chemical properties [34]. The antimicrobial effect of chitosan and chitosan derivatives was observed in a multitude of research [35,36,37]. When chitosan is dissolved, the amino groups (–NH_2_) of glucosamine units are protonated to –NH_3_^+^. The polycationic charge of chitosan is generally thought to be the most significant factor bringing about antimicrobial effects due to electrostatic interaction between the polycationic chitosan and microbial cell surface [38]. Chitosan is only active in the acidic medium; however, it was reported that the limited activity above pH 6 may originate from the poor solubility above pH 6. Chitosan was observed to be equally active against *S. aureus* at pH 5.5 and 7.2 (MIC = 256 μg/mL) [39]. Further investigations are required to fully understand the importance of the polycationic charge of chitosan.

Therefore, chemical modification of chitosan can increase the solubility of chitosan in water medium and enhance chitosan’s antimicrobial properties.

## 3. Chemical Modification of Chitosan

Modification methods primarily consist of *N*-substitution in C2–NH_2_, *O*-substitution in C6–OH, and free radical graft copolymerization [34]. Amino groups demonstrate higher reactivity than hydroxyl groups. Hence, *O*-substitution requires the protection of primary amino groups before modification [40]. There are other chemical modification methods of chitosan. For instance, K. Zhang has utilized “click chemistry” and single-electron-transfer nitroxide-radical-coupling (SET-NRC) reaction to synthesize chitosan-g-(PEO-PLLA-PEO) polymer [41]. Different methods of chemical modification define different properties. Amino groups or hydroxyl groups require protecting functional groups introduced in chitosan. The widely used protecting groups for amino groups are phthaloyl [42] and acetyl [18], while triphenylmethyl [43], trimethylsilyl [44], tertiarybutyldimethylsilyl [45], and acetyl [18]. The positive charge from the NH_3_^+^ group of the glucosamine units in chitosan and chitosan derivatives generates antimicrobial effects and reacts electrostatically with negatively charged microbial cell membranes and biofilms components [46,47]. For alleviating limitations that antimicrobial properties exist in acidic medium (pH below 6), trimethyl, 2-hydroxy-3-trimethylammoniumyl, guanidinyl, trimethyl ammmoniumyl, pyridiniumyl, and quaternary alkyl groups are introduced in chitosan to obtain a permanent positive charge, which enables chitosan to gain antimicrobial properties in the neutral and basic medium [18]. An overview of types of chemical modification is shown in Figure 2. The preparation and biological activities of chitosan and chitosan derivatives with antimicrobial properties are shown in Table 1.

### 3.1. Carboxylic Acid Chitosan Derivatives

The quaternization of chitosan is known as a method to increase solubility and positive charge [58]. The trimethylation of chitosan amino groups is a simple quaternization method [59]. Moreover, trimethyl carboxymethyl chitosan derivatives possess enhanced antimicrobial properties [60]. Carboxymethyl chitosan is a water-soluble, amphoteric chitosan derivative with nontoxicity, biocompatibility, and biodegradability, implying its great potential in medical applications [61,62,63]. Such properties are brought about by the increased surface positive charge [58]. A quaternized carboxymethyl chitosan is prepared by introducing the *N*-quaternary ammonium group via the reaction of carboxymethyl chitosan with 2,3-epoxypropyl trimethylammonium. A water-soluble chitosan methacrylate can be prepared by using K^+^(CH_3_)_3_CO^−^ dissolved in acetonitrile and methacryloyl chloride. The graft copolymer using chitosan methacrylate and different monomers (e.g., 1-vinylimidazole, methacrylamide, and 2-acrylamido-2-methyl-1-propanesulfonic acid) exhibits higher antibacterial effects than the chitosan methacrylate [64].

### 3.2. N,N,N-Trimethyl Chitosan (TMC)

*N*,*N*,*N*-trimethyl chitosan is a quaternized derivative of chitosan. Chitosan can react with methylation reagents, mostly *N*-methyl pyrrolidone (NMP), to form TMC. Physiochemically, TMC is soluble in neutral or alkaline media [53], and its hydrophilic (*N*-(CH_3_)_3_) and the hydrophobic groups (*N*-(CH_3_)_2_) makes TMC amphiphilic in nature, which is suitable for nanoparticle processing [65]. TMC can serve as an absorption enhancer affecting tight junctions [66], which facilitates the use in the drug delivery on the intestinal [67], nasal [53], and pulmonary [68]. TMC contains a permanent positive charge and good water solubility independent of pH values. The antimicrobial property of TMC against Gram-positive and Gram-negative bacteria outweighs that of chitosan, especially in alkaline medium [54]. Researchers found that TMC fibers have better absorption capacity and antibacterial properties compared with that of chitosan fibers, show nontoxicity to mouse embryo fibroblasts (MEFs) in vitro, and demonstrate potential wound healing activity in vivo [69]. TMC and heparin, an antiadhesive polymer, were deposited on modified polystyrene films to build antiadhesive and antibacterial multilayer films. The results show their antibacterial properties and antiadhesive nature to *E. coli* [70]. Vancomycin-loaded TMC nanoparticles were reported excellent antibacterial activity against *S. aureus* and were effective intracellular drug carriers due to their positive charge, suitable size distribution, sustained drug release, and substantial cell uptake [71].

### 3.3. N-(2-Hydroxyl) Propyl-3-Trimethyl Ammonium Chitosan (HTC)

HTC or HTC chloride (HTCC) is another common quaternized derivative of chitosan which also contains good aqueous solubility in acidic, neutral, and alkaline medium [55]. It is synthesized by the reaction between chitosan and glycidyl trimethyl ammonium chloride [55]. HTCC exhibits good antibacterial properties, mucoadhesive activity, and permeability enhancing activity [72]. The antibacterial activity of the polyacrylonitrile (PAN) fiber can be achieved by addition of only 1% HTCC [73]. HTCC polymers were reported to possess antibacterial activity against various drug-sensitive and drug-resistant bacteria, even the clinically isolated multidrug-resistant bacteria and pathogenic fungi. In addition, the polymers were found to kill pathogens by disrupting microbial membrane integrity. Low cytotoxic behavior against human erythrocytes and mammalian cells also exhibits their in-vitro non-toxic activity [74]. In the study on the relationship between the degree of substitution (DS) and antibacterial activity, Professor Másson and his coworkers have reported that HTC was more active than chitosan against *S. aureus*, but this activity is independent of DS, whereas the antibacterial activity of HTC decreases with an increase in DS in other cases. Additionally, a surprising result shows that only the lowest DS structure possessed more antibacterial activity than chitosan [39]. HTCC was also reported to block subsequent interaction between the S protein of coronavirus and the cellular receptor [75].

### 3.4. Hydroxypropyl Chitosan (HPC)

HPC is a water-soluble derivative of chitosan with film-forming property [48]. It is synthesized by reacting propylene epoxide with chitosan under alkaline medium [76]. HPC exhibits no inhibitory effect on *E. coli* or *S. aureus*, whereas it shows inhibitory effectiveness against the four fruit pathogenic fungi (*A. mali*, *C. diplodiella*, *F. oxysporum*, and *P. piricola*) [77]. In the study on the relationship between the degree of substitution (DS) and antibacterial activity, the antibacterial activity of HPC decreases with DS [39]. HPC was used to formulate nail lacquers containing antifungal agents such as ciclopirox for onychomycosis, a fungal infection of the nails [78]. HPC grafted to the antimicrobial peptide nisin using microbial transglutaminase (MTGase) as biocatalyst was also reported [79].

### 3.5. Thioglycolic Chitosan (TGC)

Thioglycolic chitosan is a thiolated chitosan derivative with antimicrobial properties. It is synthesized by the reacting chitosan, 1-ethyl-3-(3-dimethyl aminopropyl) carbodiimide and thioglycolic acid [39]. Recently, low molecular weight TGC has displayed excellent antimicrobial activity over the other derivatives (CMC and TMC): a 30 min treatment killed 100% *Streptococcus sobrinus* (Gram-positive bacteria) and reduced bacteria by 99.99% in *Neisseria subflava* (Gram-negative bacteria) and 99.97% in *Candida albicans* (fungi) [49]. Good mucoadhesive properties of TGC can serve as a promising tool for the mucoadhesive drug delivery systems [50]. In the study on the relationship between the degree of substitution (DS) and antibacterial activity, the degree of substitution for the TGC is very low and the antibacterial activity is similar to unmodified chitosan [39].

### 3.6. N-(2-(N,N,N-trimethylammoniumyl)acetyl)-chitin (TACin)

TACin is also a quaternized derivative of chitosan, containing quaternary *N*,*N*,*N*-trimethylammonium groups. TACin is synthesized using a combination of Boc and tert-butyldimethylsilyl (TBDMS) protection strategies [80]. This derivative was shown to have good antimicrobial activity [51] and serve as a promising permeation enhancer [52]. In the study on the relationship between the degree of substitution (DS) and antibacterial activity, the antibacterial activity of TACin increases with DS from 0.07 to 0.88. The TACin derivative with the highest DS is more active (MIC = 256 μg/mL) than TMC against *P. aeruginosa* [80].

### 3.7. Chitosan Conjugates

Some bioactive components can also be covalently linked to chitosan polymer to form antimicrobial conjugates. It was reported that an antimicrobial peptide chitosan conjugate was synthesized by the grafting of an antimicrobial peptide, anoplin, to chitosan polymers. The antimicrobial activity of the conjugate is better than anoplin [81]. Chitosan-polyethylene glycol-peptide (PEG)-peptide conjugate was reported to self-assemble into a neutral nanosphere structure, penetrating the biofilm and membrane of bacteria. Furthermore, chitosan-PEG-peptide conjugate destroys *P.*
*aeruginosa* in a planktonic form and in a biofilm form. The results indicated that chitosan-PEG-peptide conjugate can serve as antibacterial agents against pathogens combating the *P. aeruginosa* biofilm infection, which is an issue in hospitals [56].

## 4. Action Modes of Chitosan against Pathogen Microorganisms

Chitosan and chitosan derivatives exhibit different action modes towards the Gram-positive and Gram-negative bacteria. This difference in mechanisms can be attributed to the difference in the component of the cell wall. As shown in Figure 3, the cell wall of Gram-positive bacteria is composed of peptidoglycan, wall teichoic acids (WTAs) covalently linked to peptidoglycan, and lipoteichoic acids (LTAs) tied to the microorganism cell membrane [82]. WTAs and LTAs contain a negatively charged anionic backbone [82,83]. The teichoic acids can provide arranged uniform high-density negative charges in the cell wall, thereby inhibiting the passage of ions across the membrane [82].

In the case of Gram-negative bacteria, the cell envelope consists of two membranes divided by a periplasmic space comprising a thin peptidoglycan layer [84]. As shown in Figure 4, the lipid composition of the outer membrane (OM) of Gram-negative bacteria is asymmetric: the outer leaflet contains lipopolysaccharide (LPS), whereas the inner leaflet comprises a variety of phospholipids.

The surface of Gram-negative bacteria comprises negative charges from the phosphate and pyrophosphate groups of LPS in the outer layer of the OM [85]. The widely accepted mechanisms of antimicrobial effects of chitosan can be explained in four models.

### 4.1. Disrupting the Cell Membrane/Cell Wall

The first and most widely accepted model involves the electrostatic interactions between chitosan and anionic surface of Gram-positive and Gram-negative bacteria, resulting in disruption of the cell membrane [14].

In Gram-positive bacteria, positively charged chitosan can interact electrostatically with the negative charged teichoic acid in peptidoglycans, leading to destruction on the cell membrane, leakage of intracellular components, and the entrance of chitosan into the microbial cells [14,18]. A previous study confirmed the leakage of proteins and other intracellular constituents caused by chitosan [86]. Further studies showed that the hydrolysis of peptidoglycans can bring about an enhanced electrostatic interaction, which is confirmed via the assessment of electric conductivity of the bacteria mixture [87] and release of cytoplasmic β-galactosidase activity [85] from *E. coli* into the culture medium.

In Gram-negative bacteria, high negative charges given by LPS can be neutralized by positive charges from chitosan, resulting in disruption of the OM, enabling chitosan to penetrate the cell membrane, and thus lead to bacterial cell death [85].

Recently, the studies on the effect of chitosan on *E. coli* not only indicate the observation that the permeability of the OM is increased but also shows that the inner cell membrane is also damaged, resulting in leakage of the cytosolic content and microbial cell death [88,89]. Similar results are observed for the mechanism of action for antimicrobial dimethylaminoethyl-chitosan against *E. coli* and *S. aureus* [87]. Chitosan and chitosan derivative microspheres were reported to disrupt the bacterial cell membrane [86].

### 4.2. Formation of a Dense Polymer Film on the Cell Surface

High molecular weight (High-MW) chitosan can form a dense polymer film on the surface of the cell, blocking the exchange of nutrients by covering porins on the OM of Gram-negative bacteria, leading to microbial cell death [84,85]. Such profile was detected via the thicker appearance of the cell walls, implying chitosan deposition on the cell surface [85]. The flocculation effect can be detected by using a scanning electron microscope (SEM), which shows vesicle-like structures on the OM of chitosan-treated *E. coli* and *Salmonella typhimurium* [90]. However, the image showing chitosan aggregates on the OM cannot support the hypothesis. More investigations are required to evaluate the assumption.

### 4.3. Interaction with Microbial DNA

Low molecular weight (Low-MW) chitosan and chitosan hydrolysis products can penetrate the cell wall to affect DNA/RNA and protein synthesis [91]. Xing and coworkers observed the binding of oleoyl-chitosan nanoparticles (OCNPs) to DNA/RNA in the assessment of the influence of OCNPs on the electrophoretic mobility of nucleic acids [92]. The results demonstrate that the concentration of OCNPs might be positively correlated to the interactions among bacterial genomes. Moreover, the concentration of OCNPs reaching 1000 mg/L inhibits the migration of *E. coli* DNA and RNA completely. Negatively charged phosphate groups in DNA/RNA react with positively charged amino groups in OCNPs, thus inhibiting pathogens [92]. Galván Márquez and coworkers observed the inhibition of protein biosynthesis by chitosan in a test of β-galactosidase expression [93]. The antibacterial effect of chitosan oligomers on *E. coli* cells is caused by the prevention of DNA transcription caused by chitosan oligomers, which is confirmed by the detection of chitosan oligomers inside the cell using a confocal laser scanning microscope [94]. Furthermore, chitosan can inhibit mitochondrial biogenesis of *Candida albicans*, indicating that the antifungal activity of chitosan is mediated by the repression of mitochondrial function and the following ATP production inhibition [95].

### 4.4. Chelation of Nutrients by Chitosan

Metal ions (e.g., Ni^2+^, Zn^2+^, Co^2+^, Fe^2+^, and Cu^2+^) present in the bacterial surface can be chelated by amino groups of chitosan when the chelation effect overweighs the electrostatic force, namely the higher pH of the mixture than the pK_a_ of chitosan [96,97,98,99,100,101]. Divalent cations can stabilize the cell membrane of bacteria [102]. In Gram-positive bacteria, the divalent metal ions binding to wall teichoic acids (WTAs) can minimize repulsion among adjacent phosphate groups, leading to better stabilization polymer structure and the integrity of the cell wall [103,104]. Divalent cations binding to WTAs can help inhibit fluctuations in osmotic pressure between both sides of the microbial cell [105,106,107]. In the case of Gram-negative bacteria, LPS in the outer leaflet of the OM is polyanionic molecules comprising various negative charged phosphate groups [102]. The divalent metal cations can minimize the repulsive forces among aggregated negatively charged phosphate groups, maintaining the stability of the bacterial OM [102].

Chitosan contains chelating properties [108]. When the pH value of the medium is higher than the pK_a_ value of chitosan or chitosan derivatives [100], unprotonated amino groups of chitosan can donate their lone pair of electrons to the metal ions of phosphate groups in the LPS or WTAs on the cell membrane surface to form a metal complex. Positively charged amino groups of chitosan can compete with divalent cations for phosphate groups in the LPS or WTAs on the cell membrane surface [108,109]. Therefore, such a chelation reaction can lead to instability of cell surface potential and a mutual repulsion among negatively charged phosphate groups, and thus result in rupture of the microbial cell membrane [85].

In addition, it was reported that bivalent cations inhibited the activity of chitosan in the order Ba^2+^ > Ca^2+^ > Sr^2+^ > Mg^2+^. The results demonstrate that chitosan leads to the formation of “pores” in the cell membrane, and Ca^2+^ bound to the cell surface is released before the chitosan-induced leakage of the cytosolic content [110].

However, intracellular effect on DNA and RNA function or chelation of nutrients are speculative by some published work.

### 4.5. Mechanism of Antifungal Activity of Chitosan

Fungi includes the yeasts, rusts, smuts, mildews, molds, and mushrooms. Fungi, one of the most widely distributed organisms on Earth, are found in soil, water, plants and animals [111]. Many fungi were detected in the healthy human gut, mostly *Candida* yeasts. Sometimes environmental sources (e.g., molds) are likely to affect gut ecology [112].

The fungal cell wall (Figure 5) is primarily composed of chitin adjacent to the cell membrane, β-d-glucans outside the chitin fibers, and mannoproteins or mannan as the outer layer of the cell wall [113]. Chitosan contains antifungal properties on various fungal pathogens in plants and humans [114,115,116,117]. A variety of studies have clearly indicated that chitosan can bind to the phosphorylated mannosyl side in fungi, leading to disruption of the plasma membrane and leakage of intracellular materials [118,119,120]. Recently, researchers found that chitosan can also affect DNA/RNA expression and protein biosynthesis in fungi [93,121]. Xu and coworkers [122] found that fluorescently labeled oligo-chitosan localizes primarily in the cytoplasm of *P. capsici* and shows no binding to the cell wall or membrane. It is also reported that the oligo-chitosan can affect the electrophoretic mobility of the DNA and RNA, which mostly suggests that small antifungal oligo-chitosan can cross the fungal cell wall and cell membrane and bind to intracellular DNA or RNA. However, these results were contrasted by Park et al. [123], who demonstrated that oligo-chitosan (1 kD) was less effective against nine fungal strains than chitosan with higher molecular weight (3, 5, and 10 kD). Furthermore, low molecular weight chitosan and oligo-chitosan can pass through fungal cell walls to affect mitochondrial function [124].

The minimum lethal concentrations (MLCs) of chitosan against fungi are highly correlated to the molecular weight and degree of acetylation of chitosan, pH value, and the targeted fungi [117,125,126,127]. Fungicidal activity is positively associated with the degree of acetylation and negatively related to molar weight [126]. It is observed that the plasma membrane forms a barrier against chitosan in chitosan-resistant but not chitosan-sensitive fungi [128]. Chitosan-sensitive fungi have more polyunsaturated fatty acids in the cell membrane than chitosan-resistant fungi, therefore comprising more membrane fluidity and the increased negative charges as well as permeabilization [128]. Apart from the destruction of the cell wall, disruption of the cell membrane, and inhibited ribosome biogenesis, inhibiting the spore germination and mycelium growth of fungi adds to the usage of chitosan as an antifungal agent [129].

Given the negative charge of WTAs in Gram-positive bacteria, LPS in Gram-negative bacteria, and the phosphorylated mannosyl side in fungi, electrostatic reactions occur between the positively charged amino groups of chitosan and the cell surface of the pathogen microorganism. Furthermore, chitosan chelates the metal cat-ions on the surface of the bacterial membrane. High molecular weight chitosan can obstruct the exchange of nutrients on the Gram-negative bacteria. Low-molecular-weight chitosan and oligo-chitosan can inhibit DNA/RNA or protein synthesis by passing through the cell wall and cell membrane into the cytoplasm. Additionally, low-molecular-weight chitosan and oligo-chitosan result in mitochondrial dysfunction in fungi. Figure 6 exhibits the proposed modes of action of chitosan on Gram-positive, Gram-negative bacteria and fungi.

## 5. Current Treatment of the Enteric Infections

Rehydration is the primary treatment of enteric infections. Most cases can be efficiently treated with oral rehydration salts (ORS) [7]. WHO and the United Nations Children’s Fund have recommended a reduced osmolarity ORS solution (245 mOsm/L), containing 75 mmol/L of sodium, 10 mmol/L citrates, 20 mmol/L potassium, and 75 mmol/L of glucose, for reducing stool output and the incidence of vomiting [130]. Digestible food is usually recommended for patients with diarrhea [131]. Severe suffers require intravenous fluids. Apart from normal saline infusion, lactated Ringer solution is also needed [132]. Considering serum electrolyte level and urinary excretion, a potassium supplement can correct electrolyte disorders [130].

Antibiotics demonstrated benefits, including the reduction of the duration of the condition as well as the alleviation of symptoms and complications, in randomized, controlled trials (RCT) in the treatment of enteric infections [3]. In terms of *Shigella* spp., enteroinvasive *Escherichia coli*, enterotoxigenic *E. coli*, *Vibrio cholerae*, *Aeromonas*, and *Plesiomonas*, antibiotics have exhibited benefits for patients with moderate-to-severe disease in RCTs [3]. For other pathogens (e.g., *Campylobacter*), antibiotics make a modest reduction in the duration of symptoms [133]. Antibiotics are recommended for patients with the severe conditions or risk factors for severe illness, such as the elderly, pregnancies, and the immunocompromised [133]. Nevertheless, for patients with nontyphoidal *Salmonella* infection, antibiotics should not be given except in particularly severe cases, namely, in patients older than 50 years (who are at risk for a mycotic aneurysm), in infants younger than 12 months (who are at risk for *Salmonella* meningitis), in individuals with cardiac or joint prostheses, and the immunocompromised [134]. Furthermore, RCTs showed a prolonged load of the microorganisms in the stool [134]. Antibiotics for *Shiga* toxin-producing *E. coli* (STEC) infection can increase the risk for hemolytic-uremic syndrome (HUS), particularly in children younger than 10 years [135].

Moreover, antimotility agents may exacerbate the condition without effective antibiotics, resulting in absorption of enterotoxin and subsequently increased risks for HUS [3]. Considering the progressively common antimicrobial resistance against enteric infection, researchers have focused applying nonantibiotic compounds as antimicrobial agents [1]. Chitosan and chitosan derivatives showed antimicrobial activities against Gram-positive, Gram-negative, and fungi [91]. Besides such direct bactericidal and fungicidal effects, chitosan and chitosan derivatives have other mechanisms for treating enteric infections.

## 6. Antibiofilm Properties of Chitosan and Chitosan Derivatives

Biofilm is an assemblage of microbial cells imbedded in a matrix of extracellular polymeric substances (EPS) produced by microbial cells [136]. Biofilm contains extracellular polysaccharides, extracellular DNA (e-DNA), and proteins in the matrix [137,138,139]. Biofilm can serve as a defense mechanism against antibiotics and microbicides [140]. The polymeric matrix present in the biofilm impedes the access of antimicrobial compounds to the surface of bacterial cells [47]. At present, researchers concentrated on combating the pathogenesis by inhibiting biofilm formation and uprooting mature biofilm by several reactive agents [141,142,143,144,145,146,147,148]. However, several current antibiofilm agents comprise cytotoxicity properties [149,150,151,152]. Therefore, a biocompatible, biodegradable, innocuous, nonallergenic, cost-effective, and environmentally-friendly antibiofilm compound from natural origin, chitosan can be a potential antimicrobial and antibiofilm agent [153,154,155].

Antibiofilm property of chitosan is primarily attributed to the polycationic nature donated by amino groups of *N*-acetylglucosamine monomers [156,157,158]. Positive charges of chitosan can interact electrostatically with negatively charged biofilm compositions such as EPS, extracellular proteins, and e-DNA, inducing an inhibitory effect on microbial biofilm [159,160]. Thenceforth, chitosan has access to the passage through the biofilm, killing microorganisms subsequently via the above-mentioned action modes. Moreover, the conjugation of chitosan with antimicrobial peptides (AMPs) can create cationic peptide-polysaccharide to enhance electrostatic interactions between the copolymer and microbial cell membrane components as well as inhibit the growth of Gram-positive and Gram-negative bacteria [161,162]. Chitosan and chitosan derivatives in different structural forms (e.g., chemical modified ones, nanoparticles, conjugation with other polymers, conjugation with antibiofilm nanoparticles, and vehicles for drugs) exhibit antibiofilm effects against microbial biofilm [47]. Such materials in food industry or for personal use can prevent the intake of pathogens which is responsible for enteric infections, especially methicillin-resistant *Staphylococcus aureus* [163], vancomycin-resistant *Staphylococcus aureus* [164], and *Listeria* [165,166].

## 7. Gut Microbiota and Colonization Resistance against Enteric Infections

A variety of microorganisms colonize the human gastrointestinal (GI) tract, collectively termed gut microbiota, including bacteria, viruses, fungi, archaebacteria, and protozoa [167]. The majority of bacteria colonize the colon (about 10^14^) [168], which is much larger than the concentration of microbes in the stomach and the upper part of the intestine owing to the acidic gastric ambient and the rapid passage of the food through the upper GI tract [167].

Discoveries of the Human Microbiome Project (HMP) boosted the understanding of the host-microbe interactions in the intestine [169,170]. The microbiome contains 3.3 million nonredundant microbial genes, which is 150-times larger than the human genome [169]. Healthy gut microbiota is dominated by *Bacteroidetes* and *Firmicutes* [171], followed by *Actinobacteria*, *Proteobacteria*, and *Verrucomicrobia* [172], as well as methanogenic archaea (e.g., *Methanobrevibacter smithii*), eukaryote (e.g., yeasts), and various phages [173]. Environmental factors interact with endogenous factors to form individuals’ unique microbial phenotypes [174]. The gut microbiota in early life is shaped by types of delivery for pregnancy [175], host immune system [176], maternal microbiome [177], environmental microbes [178,179], as well as the solid food after birth [180]. Multiple host-endogenous and host-exogenous factors shape the gut microbiota to form a resilient and balanced gut microbiota [181]. However, once there are changes in those influencing factors, the dysbiosis can induce gut barrier dysfunction, invasion of pro-inflammatory contents (e.g., LPS), and low-grade chronic inflammation [182]. Various studies showed the association between dysbacteriosis and obesity [183], type 2 diabetes [184], and inflammatory bowel disease [185].

Healthy gut microbiota can prevent enteric infections via a variety of mechanisms, including the production of antimicrobial agents, nutrient competition, aid to intestinal mucosal barrier integrity, and immune response activation [186]. These mechanisms collectively contribute to colonization resistance (CR) against pathogens. The metabolites of the gut microbiota contain antibacterial properties, including short-chain fatty acids (SCFAs), secondary bile acids (BAs), and bacteriocins [187]. Their general modes of action are explained below.

### 7.1. Short-Chain Fatty Acids

SCFAs are metabolites from bacteria via the fermentation of nondigestible carbohydrates [188]. The SCFAs primarily include acetate, propionate, and butyrate, constituting over 90% of the total metabolites [189]. Under normal conditions, butyrate can supply energy to enterocytes and form an anaerobic milieu in the gut via β-oxidation and citric acid cycle metabolism [190]. The anaerobic environment improves the growth of obligate anaerobic bacteria (e.g., *Lactobacillus*) [187] and limits the expansion of facultative anaerobic pathogens (e.g., *Proteobacteria*) [191]. SCFAs can regulate the composition of the gut microbiota by affecting pH value and metabolic function. The concentration of SCFAs is inversely associated with pH value throughout different zones of the GI tract [192]. Butyrate contributes to the colonic mucosa’s physical and functional integrity by upregulating the expression of mucins from goblet cells in the colon [193]. Mucin, a type of mucus protein, is a major structural and functional constitute of the intestinal mucus layer [194], and the mucus generates a coating that covers the intestinal cells to protect them from exogenous and noxious substances such as pathogens and digestive enzymes [195]. Acetate can impair the metabolism of *Escherichia coli* by inhibiting the methionine biosynthesis and accumulating toxic metabolites (viz. homocysteine), reducing the growth of *E coli.* (phylum *Proteobacteria*), a marker of gut dysbiosis [191,196].

### 7.2. Bile Acids

Secondary bile acids (e.g., deoxycholic acids) possess bactericidal properties against multiple pathogens, including *Staphylococcus aureus* and *Clostridioides difficile*, disrupting cell membrane [197,198,199]. Primary bile acids (BAs) are generated in the liver and excreted in the intestinal tract to support the digestion of exogenous lipids. Afterwards, primary BAs are conjugated with glycine or taurine in the liver to increase solubility [200]. Most conjugated primary bile acids are reabsorbed in the distal ileum, while the remains are metabolized by the microbes in the colon [187]. Conjugated bile acids can be deconjugated by bile salt hydrolase (BSH) in the colon, which is profusely expressed in the gut microbiome [201,202]. The deconjugation creates two main secondary bile acids, deoxycholic acid, and lithocholic acid, via several bacteria (e.g., *Clostridium specie*) [203].

### 7.3. Bacteriocins

Bacteriocins, bactericidal peptides generated by specific bacteria, can inhibit the occupation and growth of other bacterial species [204]. The action modes include disturbing RNA and DNA function as well as disrupting the cell membrane [205]. The bacteriocins from Gram-positive bacteria are primarily produced by *Lactococcus* and *Lactobacillus* [206]. Bacteriocins from Gram-negative bacteria are usually generated by *Enterobacteriaceae* [207]. The bacteriocin Abp118, produced by *Lactobacillus*, can protect the host from the infection with *Listeria monocytogenes* [208].

### 7.4. Mucus Layer

The mucus layer is composed of inner and outer mucus layers. The inner mucus layer is consistently replenished with mucins and is anchored on the goblet cells and the intestinal epithelia [209,210,211]. It was observed that bacteria cannot penetrate the inner mucus layer due to the pore sizes down to 0.5 µm [211,212]. The outer mucus layer is four times volume of the inner mucus layer, remaining the loose netlike structure and colonized with gut microbiota [213]. At a certain distance (50 µm in mice and 200 µm in humans) from the intestinal epithelia, the inner mucus layer is changed into the outer mucus layer by endogenous proteases, creating a significantly sharp border that separates the two layers (from being attached to easily aspirated) [211,214]. The inner mucus layer can separate the intestinal epithelia and microbes in the GI tract, preventing bacterial translocation and subsequent systemic inflammation [215]. The upregulated mucins mediated by SCFAs from the healthy gut microbiota can supply energy for specific beneficial mucin-degrading bacteria (e.g., *Akkermansia muciniphila*), resulting in expanded microbiota outer mucus layer [11,215]. In summary, the mucus layer serves as the first barrier of defense against exogenous pathogens.

### 7.5. Immune Response

The intestinal immune system is significantly shaped by the gut microbiota. Immunocytes such as neutrophils and macrophages are typically the first immune barriers against infection [29]. Pathogen-associate molecular patterns (PAMPs) produced by pathogens can interact with pattern recognition receptors (PRRs), such as toll-like receptors (TLRs) and nucleotide-binding oligomerization (NOD)-like receptors (NLRs), to activate innate immunity [216,217,218]. The innate immune system includes intestinal epithelial cells, myeloid cells, innate lymphoid cells (ILCs), etc. PAMPs can stimulate TLRs and myeloid differentiation factor 88 (MyD88) expressed in intestinal epithelial cells, resulting in the secretion of IL-8 from the epithelia and the recruitment of neutrophils to the lamina propria [218,219]. In response to microbial stimuli, innate immune cells, such as dendritic cells (DC) and macrophages, secrete IL-12 and IL-18 that stimulate ILC1 to kill intracellular pathogens, as well as IL23 and IL1β that activate ILC3 against extracellular bacteria and fungi [220]. The production of IFN- γ by ILC1 can facilitate macrophage to eliminate infected cells [221]. ILC3 can secret IL-17 and IL-22 to increase the secretion of antimicrobial components, including antimicrobial peptides (AMPs) and regenerating islet-derived 3 (Reg3) family proteins, to the mucus layer in the small intestine [222,223]. IL-22 can also induce the expression of fucosyltransferase 2 (Fut2) to improve the glycosylation of the proteins expressed on the surface of enterocytes [224], which can facilitate to protect against infection by intestinal pathogens (e.g., *Salmonella enterica* serotype *typhimurium*) [225]. On the other hand, butyrate can stimulate the G-protein-coupled receptor 109A (GPR109A) on dendric cells (DCs) or GPR43 expressed on naïve T cells to inhibit histone deacetyltransferase (HDAC) function and subsequently upregulate the expression of forkhead box P3 (FOXP3), resulting in increased regulatory T cells (Treg) pool and activity [226]. The mechanism can avoid abnormal inflammation.

Immunoglobulin-A (IgA) plays an indispensable role in the mucosal defense against pathogens [227,228]. In humans and mice, over 80% of plasma cells produce IgA in mucosa-associated lymphoid tissues (MALT), whereas other IgA-producing plasma cells exist in peripheral lymphoid tissues [229]. Gut microbiota can utilize IgA for colonization resistance against exogenous pathogens and commensal-bacteria translocation [230,231]. SCFAs can promote retinoic acid production from DCs, which in turn promotes differentiation of IgA-producing plasma cells [232]. In the colonic lamina propria, IgM-expressing B cells convert to IgA-producing B cells at this site, and the antibody product convert to IgA2 from IgA1. Following the activation of TLRs on the intestinal epithelia, intestinal epithelial cells can secret thymic stromal lymphopoietin (TSLP) to enhance the production of a proliferation-inducing ligand (APRIL), B cell-activating factor (BAFF), and nitric oxide (NO) by DCs, resulting in increased IgA-producing plasma cells as well [233]. IgA in responses to pathogens is T-cell-dependent and is thought to induce high-affinity IgA against pathogens, termed ‘classical’ IgA [234]. Besides, the most part of commensal bacteria is coated with low-affinity, ‘innate’ IgA [235,236]. Classical IgA and innate IgA combine to establish the overall IgA pool in the intestine to facilitate tolerating a complex gut microbiome and prevent enteric infections [230,233]. As shown in Figure 7, gut microbiota plays an integral role in resisting the colonization of intestinal pathogens.

## 8. The Role of Chitosan in the Treatment of Enteric Infections

In the treatment of enteric infections, chitosan, oligo-chitosan, and their derivatives can serve as antimicrobial drug delivery vehicles, as prebiotics to improve the colonization resistance against enteric infections, and as antimicrobial agents independently, and conjugate with other reactive agents to increase antimicrobial activities.

### 8.1. Chitosan as Drug Delivery System

Apart from the antimicrobial properties, chitosan can increase the pH sensitivity of the drug release of antimicrobial agents, enabling intestine-targeted antimicrobial effects [237]. Formulated chitosan-coated amphotericin-B-loaded nanostructured lipid carriers (ChiAmp NLC) can prevent Amphotericin-B from exposure outside the intestine, decrease toxicity to enterocytes and erythrocytes, and thus enhance the bioavailability of Amphotericin-B [237]. Bovine serum albumin (BSA)-chitosan core (CS)-nano-delivery-systems (BSA-CS-NDS) enables the effective delivery of carvacrol, a natural antimicrobial agent, to the intestine for successful removal of *Salmonella enterica* [238]. Carla Mura and her coworkers prepared two chitosan amide conjugates of metronidazole, metronidazole-glutaryl- and metronidazole-succinyl-chitosan conjugates. The results have shown adequate stability of the two conjugations in the acidic environment as well as a potential as colon-targeted delivery systems of metronidazole [239]. Chitosan nanoparticle intracellular delivery system of ceftriaxone sodium can reduce the count of *Salmonella typhimurium* in intestinal cells and macrophages [240]. Chitosan- coated alginate microparticle system of lactoferrin, a protein delaying *Clostridioides difficile* growth and inhibiting toxin production, can assist in the stability of lactoferrin and protection from *C. difficile*-induced intestinal epithelial damage [241]. Albendazole-associated chitosan nanoparticles (ABZ-CS-NPs) can improve the stability of albendazole in acidic ambient and absorption of albendazole in the intestine, suggesting improved effects of killing enteric parasites [242]. Chitosan nanoparticles can serve as carriers for supernatant of mesenchymal stem cells for the treatment of multidrug-resistant (MDR) *Vibrio cholerae* infections [1].

### 8.2. Chitosan as Antimicrobial Agents

Chitosan oligosaccharide sensitizes multidrug resistant *Staphylococcus aureus* to antibiotic formulations by electrostatically interacting with multidrug efflux pumps [243]. Chitosan nanoparticles can serve as a good candidate among natural giardiacidal agents [244]. Mohamed Mammeri and his coworkers have observed the anti-cryptosporidium properties of chitosan in vitro and in vivo [244]. Chitosan and chitosan nanoparticles were observed to contain antimicrobial activity against gastrointestinal pathogens such as *Salmonella* spp. and *E. coli* [245]. Chitosan nanoparticles can facilitate the inhibition of norovirus, the most frequent cause of nonbacterial diarrhea [246]. A study investigated the potential effect of chitosan particles to enhance the immune response against *Hymenolepis nana*, the most common intestinal cestode [247]. Moreover, Aleksandra Milewska and her coworkers have prepared a cationically modified chitosan, *N*-(2-hydroxypropyl)-3-trimethylammonium chitosan chloride (HTCC), which can be utilized as potential inhibitors of currently circulating highly pathogenic coronaviruses, namely severe acute respiratory syndrome coronavirus 2 [SARS-CoV-2] and Middle East respiratory syndrome coronavirus [MERS-CoV] [248].

### 8.3. Chitosan Conjugation with Other Polymers or Nanoparticles

As chitosan can reduce the numbers of *E. coli* O157:H7 in feces, remain nontoxic to host, and possess antimicrobial properties against *E. coli*, antibody-conjugated chitosan nanoparticles are utilized to selectively kill *Shiga* toxin-producing *Escherichia coli* (STEC) without inhibiting the growth of beneficial bacteria [249]. Cranberry proanthocyanidin-chitosan composite nanoparticles (PAC-CHT NPs), which are formulated using 10:1 to 30:1 proanthocyanidin to chitosan weight ratio, can form stable and bioactive nanoparticles for potential applications in the treatment of pathogenic *Escherichia coli* infection [250]. Ziyin Cui et al. formulated mannose-modified chitosan microspheres conjugating with mucosal vaccines against *Pseudomonas aeruginosa* infection in the intestine [251]. Preparation methods and biological activities of chitosan conjugation with other polymers and nanoparticles are shown in Table 2.

### 8.4. Chitosan as Prebiotics to Improve Colonization Resistance against Enteric Pathogens

Chitosan and chitosan derivatives can be fermented by the intestinal microbiota, and the metabolites such as short-chain fatty acids (SCFAs) are capable of increasing the growth of probiotics [255,256] (e.g., *Bifidobacterium* spp. and *Lactobacillus* spp.) and the exclusion of pathogens (e.g., *Streptococcus mutans* [257], *E. coli*, *Shigella dysenteriae*, *Aeromonas hydrophila*, *Salmonella typhimurium* and *Bacillus cereus* [28]).

Furthermore, chitosan can serve as prebiotics to inhibit or prevent the growth of harmful bacteria by producing SCFAs and other beneficial metabolites [187]. The improved gut microbiota can prevent the infection of conditional pathogens such as *Clostridium difficile.* In the treatment of *Shiga*-toxin-producing *E. coli* (STEC), acetate was more effective in inhibition of STEC than butyrate and propionate [258], and butyrate can improve STEC bacterial clearance [259]. One study has investigated the effects of SCFAs on *Yersinia enterocolitica* at 4 °C. Propionic acid is similarly effective in inhibiting the growth of *Yersinia enterocolitica* with anaerobic and aerobic culture methods [260]. However, *V. cholerae* uses a wide variety of mechanisms to overcome colonization resistance. *V. cholerae* is capable of using its acetate switch, the shifting from elimination to assimilation of acetate, to increase its virulence [261].

## 9. Conclusions and Outlooks

Ingestion of pathogenic microorganisms and the disruption of gut microbiota lead to enteric infections. In light of dehydration and inflammatory response, rehydration and antibiotic therapy are essential to the treatment of enteric infections. Nevertheless, antibiotic-induced drug resistance and gut dysbiosis has led to growing attention to the use of nonantibiotic nontoxic antimicrobial agents as alternatives for treatment and disinfection. Chitosan, a biocompatible, nontoxic and biodegradable polysaccharide from natural origin, is approved generally as safe by the United States FDA. Chitosan and chitosan derivatives can kill pathogenic microorganisms by neutralizing negative charges on the microbial surface. Besides, chemical modifications give chitosan derivatives better water solubility and antimicrobial property. This review gives a summary of the preparation of chitosan, its derivatives, and the conjugates with other polymers and nanoparticles. Furthermore, we summarize chitosan derivatives with antimicrobial properties: carboxylic acid chitosan derivatives, *N*,*N*,*N*-trimethyl chitosan, *N*-(2-hydroxyl) propyl-3-trimethyl ammonium chitosan, hydroxypropyl chitosan, thioglycolic chitosan, *N*-(2-(*N*,*N*,*N*-trimethylammoniumyl)acetyl)-chitin. Additionally, we elucidate the direct action modes of chitosan: positively charged amino groups from chitosan can disrupt the cell membrane/wall by electrostatically interacting with negative charged constitutes on the microbial cell surface; high-MW chitosan can bind to porins on the OM of Gram-negative bacteria to block the exchange of nutrients, leading to cell death; low-HW chitosan can pass through the cell wall to affect the biogenesis of DNA/RNA and protein; unprotonated amino groups of chitosan can chelate divalent metal cations on the cell surface to destruct cell walls or membranes. We illustrate indirect mechanisms of antimicrobial chitosan, viz. inhibition of biofilm facilitating contact of chitosan with microbial cells and regulation of gut microbiota enhancing colonization resistance against pathogens. In addition to the summary of current treatment for enteric infections, we conclude the role of chitosan and chitosan derivatives in the antimicrobial agents in enteric infections, viz. chitosan serving as antimicrobial agents, drug delivery carriers for antimicrobial agents, and prebiotics to enhance colonization resistance against pathogens. Chitosan can conjugate with other reactive components as antimicrobial agents as well.

Currently, chitosan is approved by GRAS of FDA, and there are several antimicrobial dressings and drug delivery vehicles using chitosan and chitosan derivatives are approved by FDA. However, they are approved new drug application (NDA), not approved drugs. Although substantial studies of safety and toxicological are available, research for the mutagenicity and genotoxicity of chitosan are insufficient, which are the FDA may require for additional approval. In addition to limited in vivo research, current chitosan-based antimicrobial agents undergoing clinical trials are mostly for external use, not oral pharmaceuticals. Therefore, antimicrobial chitosan requires in vivo research and clinical trials, and the genotoxicity of chitosan requires further investigation. Our hope is to develop more useful and safe chitosan derivatives and conjugates to improve the clinical treatment of enteric infections.

## Figures and Tables

**Figure 1 molecules-26-07136-f001:**
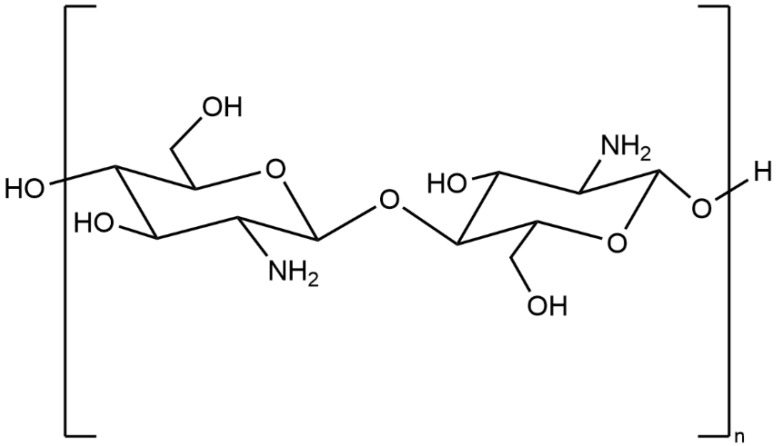
Structure of chitosan.

**Figure 2 molecules-26-07136-f002:**
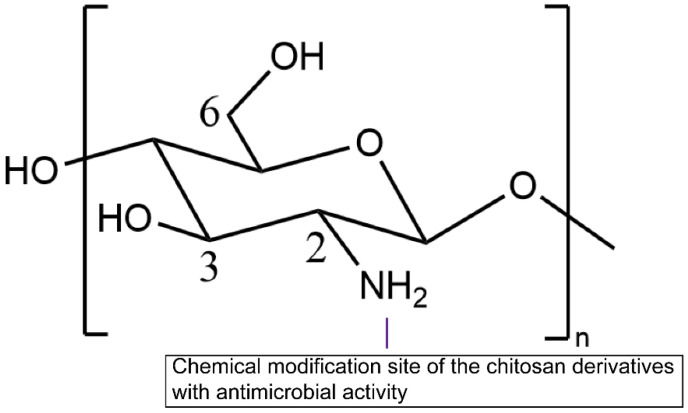
Schematic diagram of chitosan chemical modification.

**Figure 3 molecules-26-07136-f003:**
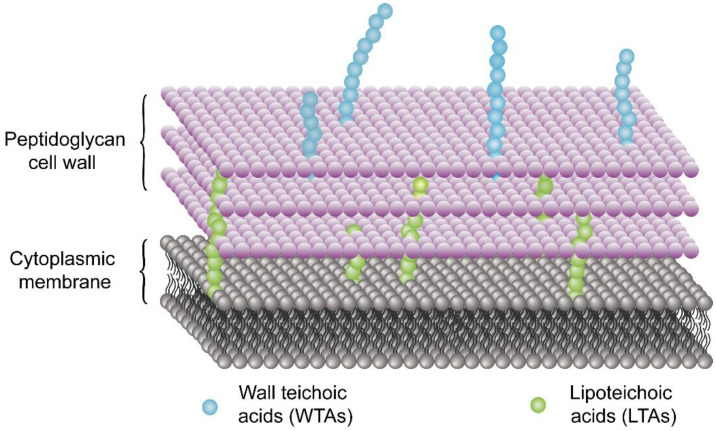
Teichoic acid polymers are located within Gram-positive cell wall.

**Figure 4 molecules-26-07136-f004:**
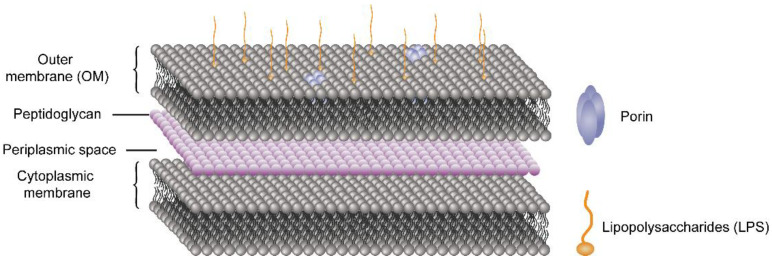
Cell envelope of Gram-negative bacteria.

**Figure 5 molecules-26-07136-f005:**
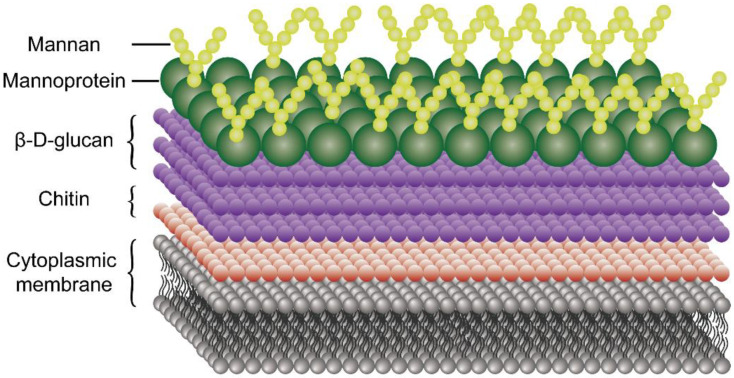
Schematic overview of fungal cell wall structure.

**Figure 6 molecules-26-07136-f006:**
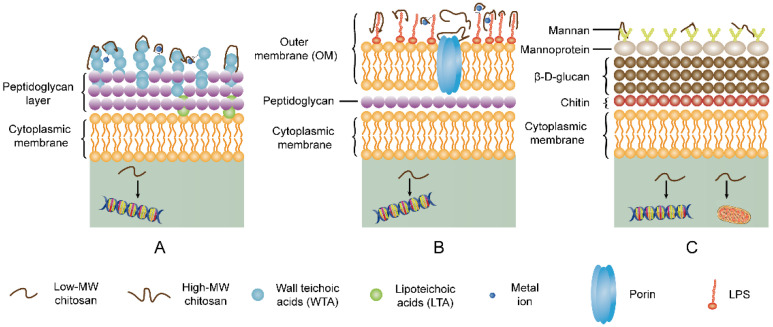
Schematic overview of action modes of chitosan on pathogen microorganisms. (**A**) Gram-positive bacteria, (**B**) Gram-negative bacteria, and (**C**) fungi. Owing to negative charges of WTAs in gram-positive bacteria, LPS in gram-negative bacteria, and phosphorylated mannose in fungi, positively charged chitosan are neutralized by above-mentioned components and induce leakage of intracellular components. Moreover, chitosan chelates metal cations on surface of bacteria, resulting in rupture of microbial cell membrane. High-molecular weight (MW) chitosan hinders exchange of nutrients by binding to porins on OM of Gram-negative bacteria, and thereby leading to bacterial cell death. Low-molecular weight (MW) chitosan can inhibit DNA/RNA or protein biosynthesis after penetrating into cytoplasm. Additionally, low-MW chitosan can induce mitochondrial dysfunction and reduced ATP production.

**Figure 7 molecules-26-07136-f007:**
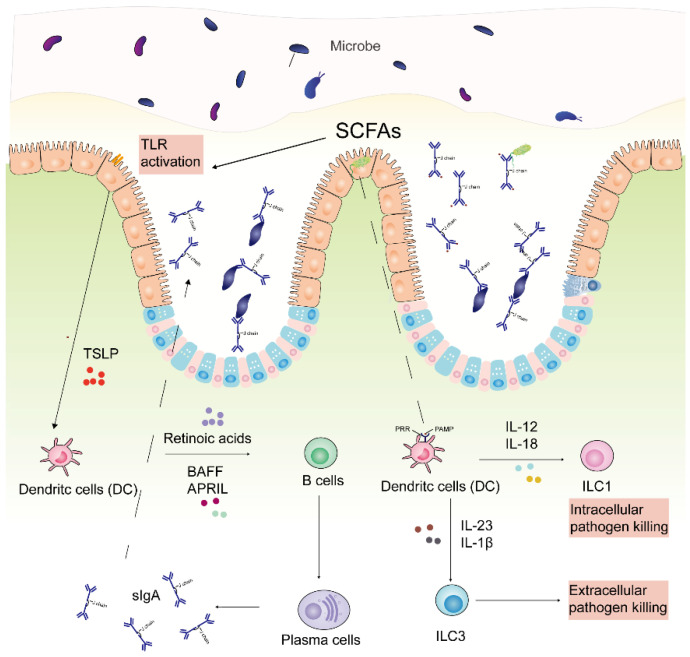
Immune response and colonization resistance.

**Table 1 molecules-26-07136-t001:** Table showing preparation methods and biological activities of chitosan and its derivatives.

Polymer	Preparation	Biological Activities	Citation
Hydroxypropyl chitosan	Reacting with propylene epoxide under alkaline medium (NaOH)	Water solubility, film-forming property, antibacterial property	[39,48]
Thioglycolic Chitosan	Reacting with 1-ethyl-3-(3-dimethyl aminopropyl) carbodiimide and thioglycolic acid	High antimicrobial property and good mucoadhesive property	[49,50]
*N*-(2-(*N*,*N*,*N*-trimethylammoniumyl)acetyl)-chitin	A combination of Boc and tert-butyldimethylsilyl (TBDMS) protection strategies	Antimicrobial property and enhancing permeation	[51,52]
Carboxymethyl chitosan	Reacting with 2-chloroacetic acid with NaOH	Enhanced antimicrobial property and water solubility	[39]
*N*,*N*,*N*-trimethyl chitosan	Reacting with methylation reagents	Antimicrobial property and enhanced solubility in alkaline medium	[53,54]
*N*-(2-Hydroxyl) Propyl-3- Trimethyl Ammonium Chitosan	Reacting with glycidyl trimethyl ammonium chloride	Antimicrobial property and good aqueous solubility in acidic, neutral, and alkaline medium	[39,55]
Chitosan- polyethylene glycol-peptide (PEG)-peptide conjugate	The chitosan was PEGylated by a carboxyl and azideterminated polyethylene glycol; peptide was conjugated onto chitosan-PEG through a click reaction	Antimicrobial property and antibiofilm activity against *Pseudomonas aeruginosa*	[56]
Thiosemicarbazone *O*-Carboxymethyl-chitosan (TCNCMCs) derivatives	Preparation of *O*-Carboxymethyl chitosan (CMCs): alkalized chitosan reacting with monochloroacetic, acetic acid, and methanol; TCNCMCs preparation: CMC reacting with ammonium hydroxide and carbon disulfide, followed by adding sodium chloroacetate and hydrazine hydrate	High antimicrobial and antifungal properties	[57]

**Table 2 molecules-26-07136-t002:** Table showing preparation methods and biological activities of chitosan conjugation with other polymers and nanoparticles.

Chitosan Conjugation with Other Polymers and Nanoparticles	Preparation	Biological Activities	Citation
Chitosan coated PLA (poly d, l-lactic acid) nanoparticles	Coated on the surface of PLA nanoparticles which are prepared by nanoprecipitation method	High cornea permeation and high sustained release of 5-FU in conjunctival/corneal squamous cell carcinoma	[252]
Antibody-conjugated chitosan nanoparticles	Preparation of chitosan nanoparticles (CNs): chitosans are dissolved in 2% acetic acid and mixed with 1% Tween 80, followed by the addition of a 10% sodium sulfate solution and centrifugation at 8200 *g*; Bioconjugation of IgY antibodies to CN: The CNs are dissolved in 0.1 M sodium acetate buffer, followed by an addition of antibodies, 1-Ethyl-3-(3-dimethylaminopropyl)carbodiimide (EDC), sulfo-*N*-hydroxysulfosuccinimide, and the centrifugation at 39,800 *g*	Enhanced and specific antimicrobial activities against Shiga toxin-producing *Escherichia coli* (STEC)	[249]
Chitosan-based nanocomposites	Chitosan prepared in acetic acid, silver, and copper nanoparticles are dispersed in ethanol by sonication and precipitated in an alkaline medium.	Increased antimicrobial properties	[253]
Cranberry proanthocyanidins-chitosan composite nanoparticles (PAC-CHT NPs)	Chitosan (CHT) is prepared in 0.5% acetic acid, filtered and degassed, followed by linking to cranberry proanthocyanidins (PAC) through hydrogen bonding.	Higher bioactive than CHT and PAC alone. Increased bioactivity of PAC-CHT NPs against *E. coli.*	[250]
Antibody-loaded-mannose-modified chitosan microspheres	Mannose-modified chitosan (MC) preparation: dissolved chitosan is treated with mannose and sodium cyanoborohydride; chitosan microsphere preparation: sodium tripolyphosphate (TPP) solution is added dropwise to MCs under 15 W sonication; antibody-loaded chitosan microsphere preparation: dispersing 5 mg of antibodies in 1.0 mL of phosphate-buffered saline (PBS) containing 30 mg of microspheres.	Mannose-modified chitosan microspheres can serve as a promising subunit delivery system for vaccines against *P. aeruginosa* infection.	[251]
Chitosan-Caffeic Acid Conjugate	Chitosan is dissolved in 2% acetic acid and reacts with 1.0 M hydrogen peroxide containing ascorbic acid. Caffeic acid is added to the mixture for 24 h at room temperature.	Antibacterial activity of against acne-related bacteria	[254]

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
