# Peer review of "Antimicrobial Properties of Chitosan and Chitosan Derivatives in the Treatment of Enteric Infections"

_molecules, 2021, doi:10.3390/molecules26237136_

Round 1

Reviewer 1 Report

Manuscript molecules-1440813: Antimicrobial Properties of Chitosan and Chitosan Derivatives in the Treatment of Enteric Infections of Dazhong Yan, Yanzhen Li, Yinling Liu, Na Li, Xue Zhang, Chen Yan.

The manuscript is an exhaustive treatment of the antimicrobial activity of chitosan and its derivatives. It ranges adequately from chemical to biological aspects. It is very well written, and the work is well organized.

The work can be accepted for publication in the newspaper but with less revision.

Paragraph 3 lacks an illustrative description of the methods used to obtain synthetic derivatives.

In my opinion, it is necessary to insert a figure with the structures of the derivatives, an example by type of substitution (from 3.1 to 3.5) or a figure with an exemplary summary for each paragraph. I think that the chemical part should be more emphasized since it is an integral part of the study.

Pay attention to the coefficients of functional groups, they must be subscript

Author Response

Thank you very much for your kind suggestions on our manuscript entitled " Antimicrobial Properties of Chitosan and Chitosan Derivatives in the Treatment of Enteric Infections" (ID:1440813). Those comments are valuable and helpful for revising and improving our paper. According to the comments of you, we have revised our manuscript as follows:

>>> Answer to the comments of reviewer #1:

Question one: Paragraph 3 lacks an illustrative description of the methods used to obtain synthetic derivatives.

Answer one: According to the reviewer’s suggestion. We have added a description of the methods used to obtain chitosan from chitin in lines 47 – 52.

Question two: it is necessary to insert a figure with the structures of the derivatives, an example by type of substitution (from 3.1 to 3.5) or a figure with an exemplary summary for each paragraph

Answer two: We appreciate the reviewer’s kind suggestion. We have inserted a figure of an overview of chitosan chemical modification to summarize types of substitution (from 3.1 to 3.5) in lines 238-241.

Question three: Pay attention to the coefficients of functional groups, they must be subscript.

Answer three: We appreciate the reviewer’s kind suggestion. We are sorry for our negligence. We have altered coefficients of functional groups into subscript (red mark).

Special thanks for the good comments.

Best regards,

Chen Yan

Reviewer 2 Report

 Dazhong et al., report   “Antimicrobial Properties of Chitosan and Chitosan Derivatives  in the Treatment of Enteric Infections” Manuscript needs some modification and addition as suggested below:

Comments

1          Author should insert a table summarizing chitosan and its derivatives, preparation method, their   biological activities along with references.

2          Author also should insert another table showing chitosan Conjugation with Other Polymers or Nanoparticles and their biological activities along with references.

3          On page 8, Mechanism of Antifungal Activity of Chitosan is described, however, no         information in the introduction about fungi.

4          C2–NH2 should be corrected as C2–NH2  throughout the manuscript.

5          NH3+ should be corrected as NH3+ .

6          K+(CH3)3CO- should be corrected as K+(CH3)3CO-

7          Abstract and conclusion should be revised according to new addition.

Author Response

Thank you very much for your kind suggestions on our manuscript entitled " Antimicrobial Properties of Chitosan and Chitosan Derivatives in the Treatment of Enteric Infections" (ID:1440813). Those comments are valuable and helpful for revising and improving our paper. According to the comments of you, we have revised our manuscript as follows:

>>> Answer to the comments of reviewer #1:

Question one: The author should insert a table summarizing chitosan and its derivatives, preparation

method, their biological activities along with references.

Answer one:  We appreciate the reviewer’s kind suggestion. We have added a table of the preparation methods and biological activities of chitosan and its derivatives along with references in lines 243-245.

Question two: The author also should insert another table showing chitosan conjugation with other

polymers or nanoparticles and their biological activities along with references.

Answer two:  We appreciate the reviewer’s kind suggestion. We have inserted a table showing preparation methods and biological activities of chitosan conjugation with other polymers or nanoparticles along with references in lines 650-652.

Question three: On page 8, the mechanism of antifungal activity of chitosan is described, however, no

information in the introduction about fungi.

Answer three: We appreciate the reviewer’s kind suggestion. We have added an introduction of fungi in lines 336-339.

Question four: C2–NH2 should be corrected as C2–NH2 throughout the manuscript.

Answer four: We are sorry for our incorrect writing. We have corrected C2–NH2 to C2–NH2 in lines 115, 118, 128, and 147.

Question five: NH3+ should be corrected as NH3+

Answer five: We are sorry for our incorrect writing. We have corrected NH3+ to NH3+ in lines 121 and 138.

Question six: K+(CH3)3CO- should be corrected as K+(CH3)3CO-

Answer six: We are sorry for our incorrect writing. We have corrected K+(CH3)3CO- K+(CH3)3CO- in line 206.

Question seven: Abstract and conclusion should be revised according to the new addition.

Answer seven: We appreciate the reviewer’s kind suggestion. We have revised the abstract and conclusion according to the new addition in lines 15 and 665-666 (red mark).

Special thanks for the good comments.

Best regards,

Chen Yan
